# A Comparison of Acute Effects of Climbing Therapy with Nordic Walking for Inpatient Adults with Mental Health Disorder: A Clinical Pilot Trial

**DOI:** 10.3390/ijerph19116767

**Published:** 2022-06-01

**Authors:** Lisa Thaller, Anika Frühauf, Alexander Heimbeck, Ulrich Voderholzer, Martin Kopp

**Affiliations:** 1Department of Sports Science, University of Innsbruck, 6020 Innsbruck, Austria; anika.fruehauf@uibk.ac.at (A.F.); martin.kopp@uibk.ac.at (M.K.); 2Schoen Clinic Roseneck, 83209 Prien am Chiemsee, Germany; a.heimbeck@schoen-klinik.de (A.H.); u.voderholzer@schoen-klinik.de (U.V.); 3Department of Psychiatry and Psychotherapy, University Hospital of Munich (LMU), 80336 Munich, Germany

**Keywords:** climbing therapy, affective state, anxiety, self-efficacy, depression, obsessive–compulsive disorder

## Abstract

As climbing therapy is increasingly used for mental health disorders, the present study aimed to compare acute effects of a therapeutic climbing intervention (CT) on affective responses, anxiety, and self-efficacy with those of Nordic walking (NW) and a sedentary control condition (SC) in an inpatient setting with persons with depression, anxiety, or obsessive–compulsive disorders. A total of 21 inpatients (32 ± 12.2 years) participated in a clinical trial in all interventions using an experimental within-subject design. Anxiety and self-efficacy were assessed preintervention (t_0_) and postintervention (t_2_) using the State-Trait Anxiety Inventory and the General Self-Efficacy Scale, and affective responses were additionally evaluated during (t_1_) and 180 min after the intervention (t_3_) using the Feeling Scale, Felt Arousal Scale, and Positive and Negative Affect Schedule. Statistical evaluation was performed with a 3 × 2 or 3 × 4 repeated measures ANOVA. Significant interaction effects were found for affective responses regarding positive affect, affective valence, and perceived activation (*p* < 0.015) favoring CT over NW and SC. For anxiety, a significant interaction effect was found (F(2.40) = 6.603; *p* = 0.003; η^2^ = 0.248), and also perceived self-efficacy increased significantly (F(2.40) = 6.046; *p* = 0.005; η^2^ = 0.232). Single CT sessions may enhance affective responses and self-efficacy and reduce anxiety in inpatients with mental health disorders to a higher extent than NW. CT as part of an inpatient therapy program may help to improve key affective mechanisms and should be further studied in comparison with other exercise interventions with comparable intensity.

## 1. Introduction

Mental and behavioral disorders have grown considerably within the last years [1,2,3,4]. Mental disorders are associated with a great burden of disease not only individually but also societally [5]. Regular physical activity (PA) decreases mortality risk and is positively associated with mental health [6,7,8,9]. PA has been used as a form of therapy in the treatment of mental health disorders; it decreases symptom severity and can be understood as part of behavioral activation therapy [7,9,10,11,12,13]. Many studies show that exercise has a positive effect on symptoms, affective state, well-being, and quality of life [8,14,15,16].

Various studies show that PA improves the affective state in people with mental health problems, which can be seen as a short-term regulation of state of mind [15,17,18]. PA also has an impact on state anxiety and self-efficacy [19,20,21]. The relationship between PA and self-efficacy is very complex, and most studies discuss the influence of self-efficacy, defined as a person’s belief in own capabilities to perform a specific action required to attain a desired outcome, on exercise behavior and not the other way around [22,23]. Only few studies have explored the effect of PA on global self-efficacy [22]. The authors postulated that self-efficacy was increased by the mastery experience in physical activity. According to the Exercise and Self-Esteem Model (ESEM), self-efficacy is reflected in global self-esteem [24,25]. Furthermore, a group of researchers found that even a single session leads to an increase in self-esteem [26,27]. Initial studies show that short-term situation-specific changes can be measured [28,29,30].

However, taking advantage of the positive effects of physical activity in general, climbing therapy (CT) for people with mental or behavioral disorders has been established for several years in clinical settings [31,32,33,34]. CT is widely used but only partially researched in the field of psychosomatics, and there are no studies investigating the effects of climbing therapy on mental-health-related outcomes in an inpatient setting with adults yet [32]. Some studies have examined the efficacy of CT over a multiple-week intervention (>6 weeks) in outpatients with mental health disorders compared with a waitlist control group with the main outcome of symptom change [33,34,35,36]. Climbing seems to encompass and address many different aspects with the potential to change in mental health in the therapeutic setting: Anxiety, attention regulation, social relations, communication, joy and flow experience, borders, self-esteem, and self-efficacy, and so on [37]. Only a few aspects have been empirically investigated, and studies show increased positive affective states, decreased anxiety, and increased self-efficacy in the CT groups compared with different control groups and conditions [33,34,35,38,39,40]. In a recent study, Luttenberger et al. [31] found bouldering psychotherapy to be as effective as cognitive behavioral therapy. Those results were seen after 10 weeks of intervention and maintained up to 1 year after intervention [31].

Although scientific research regarding CT in patients with mental health disorder is still very limited [32], recent studies investigating the effects of therapeutic climbing have shown improved mental health and changes in anxiety, affective responses, and self-efficacy in CT compared with a control group [33,35,40,41]. Luttenberger and colleagues even postulated that climbing in an outpatient setting has the same effect as cognitive behavioral therapy [31]. To further strengthen the importance of therapeutic climbing compared with other sports in the multimodal therapy approach, CT will be compared with another endurance sport in the following: Nordic walking (NW) is a recommended aerobic exercise for patients with mental disorders [42]. An improvement of the current mental state and mood-lifting effects through NW have been observed frequently in depressed persons [43,44,45]. Since CT is characterized by a higher economic cost compared with most PA group therapies (e.g., infrastructure and equipment, high staff-to-patient ratio), Frühauf and colleagues call for a randomized controlled trial with the three interventions—climbing, aerobic exercise, and social contact group—in a homogeneous group of patients [32]. To be able to place the effectiveness of climbing in the context of exercise therapies, climbing should be compared with another PA. So far, only the long-term and not the acute effects immediately after the intervention have been evaluated, which is why the present study aims to assess acute changes of psychological variables through CT in comparison with an aerobic PA and a control group. Acute effects are of particular interest in the inpatient setting, as the change can be predominantly attributed to the intervention in question. The importance of affective responses to therapy adherence and PA behavior also reinforces the need to examine acute effects [40,46]. To the best of our knowledge, no study has compared CT, aerobic physical activity, and a sedentary control group with respect to acute affective responses, state anxiety, and perceived self-efficacy in an adult inpatient setting yet. Our research group, therefore, hypothesized that therapeutic climbing would have a greater impact on affective responses, state anxiety, and perceived self-efficacy than an endurance sport or a sedentary control condition. Thus, the present study aims at answering the following research question: does climbing therapy lead to a positive change in the three constructs’ affective responses, state anxiety, and perceived self-efficacy compared with the control group’s physical activity and sedentary control condition?

## 2. Materials and Methods

The trial was designed as an experimental within-subject study with repeated measures and counterbalancing of participants. All participants attended three treatment sessions (climbing therapy, Nordic walking, sedentary control condition). The study was conducted following the Ethical Guidelines of the Declaration of Helsinki. All interventions were part of the regular inpatient therapy program. Written informed consent was obtained from each participant prior to the evaluation of the activities. The procedure was approved by the clinical direction and the Institutional Review Board for Ethical Questions in Science of the University of Innsbruck.

### 2.1. Participants

Participants were inpatients at a psychosomatic clinic in Germany (*n* = 21, 32 ± 12.2 years, 13 female and 8 male participants). The inclusion criteria for the study were people who underwent inpatient treatment at the clinic and had anxiety or obsessive–compulsive disorder or depression according to ICD-10 as their main diagnoses. The main diagnoses resulted from the clinic-specific climbing therapy concept, where priority is given to patients with the above-mentioned diagnoses. In addition, they had to be over 18 years old to participate voluntarily in the study and to be able to speak German. Exclusion criteria were defined considering health and safety aspects: body weight of over 120 kg, acute psychosis, acute suicide risk, acute cardiovascular complaints, explicit orthopedic problems, and other medical contraindications to sport. 

### 2.2. Procedure

Due to the within-subject design, all participants took part in the three interventions (1) climbing therapy (CT), (2) Nordic walking (NW), and (3) sedentary control condition (SC). All the three interventions are part of the standard clinical treatment offered to patients, and the effects were measured within the same week with 1-day break in between to reduce other therapy effects (e.g., change in pharmacotherapy, effects of group therapy) and to avoid carryover effects—for example, CT on Monday, NW on Wednesday, and SC on Friday. The starting order of the interventions varied through the weekly treatment plan of the patients and to avoid data bias by always using the same order (see Figure 1).

Measurements for the two exercise interventions were carried out in the third or fourth session of each intervention, so the participants were already used to the intervention. The time for the respective conditions was comparable: for the CT with 100 min, a slightly longer period was scheduled compared with 75 min of NW or SC due to the necessary safety information and discussion regarding goal setting, which made the exercise intervention time comparable. Data collection took place at 4 time points: at the beginning (t_0_), during the intervention (t_1_), immediately after the intervention (t_2_), and 180 min after the intervention (t_3_) (see Figure 2).

At each time point, different questionnaires related to the three constructs affective responses, anxiety, and self-efficacy were evaluated. Data collection was mostly performed with tablets, and only during the NW intervention and 180 min after the intervention, the questionnaires were filled out in a paper pencil format. Data collection was carried out by the respective sports therapists, who were not blinded. Partly the therapists differed between the interventions, but not always for planning reasons on the part of the clinic.

### 2.3. Interventions

#### 2.3.1. Climbing Therapy

The climbing intervention consisted predominantly of rope climbing at the clinic’s climbing wall. Group size was six patients and one sports therapist with additional training as climbing coach. Questionnaires were filled out after a short introduction by the sports therapist. Afterward, the group focused on the current well-being and discussed today’s goals. Next, teams of three patients started to climb. Because all participants learned the technique of belaying in the second session of the climbing group, they could belay each other during the intervention. Depending on their own goals, they climbed and belayed and tried different exercises, such as climbing blind or practicing falling. Each participant was able to climb between three and five times. At the end, a reflection round was held, where the patients could discuss the achievement or nonachievement of their goals and make a transfer to everyday life. After that, the data were collected.

#### 2.3.2. Nordic Walking

The NW intervention as an aerobic endurance sport was chosen to control for group and PA effects. Group size differed between 8 and 12 patients and one sports therapist. After a short warming up with an explanation of the technique, the patients, accompanied by one sports therapist, walked with sticks on paths outside the hospital, along the lakeside or through the nearby forest. Patients were allowed to walk next to each other if they wanted to make contact. At the end, stretching exercises were briefly performed in front of the clinic. The assessment of the dependent variables took place afterwards.

#### 2.3.3. Sedentary Control Condition

The sedentary control condition (SC) was established as a control condition close to everyday life in a stationary setting. The participants were allowed to engage in their usual activities in the clinic (except PA). The SC intervention can be equated with a waitlist control condition. For traceability, the activities were documented afterward, with attention to caffeine and nicotine consumption. Questionnaires were filled out before, during, and after the intervention with tablets next to the office of sports therapy. 

### 2.4. Measurements

#### 2.4.1. Affective Responses

Affective responses were assessed with three different questionnaires to identify affective valence, perceived activation, and positive and negative affect.

##### Affective Valence

Affective valence was recorded by the Feeling Scale (FS) [47]. It is a single-item rating scale from “very bad” (−5) to “very good” (+5), whereas zero is “neutral”. The participants were asked, how good or bad they feel at the moment. Convergent validity for the FS was demonstrated previously, also in the context of PA, for the English version (r = 0.51 to 0.88) and the German version (r = 0.72 to 0.73) [47,48,49]. FS has been applied in combination with physical activity in several studies [46,49,50].

##### Perceived Activation

Perceived activation was evaluated by the Felt Arousal Scale (FAS) [51]. Participants should answer how activated they currently feel on a single-item 6-point rating scale from “low arousal” (+1) to “high arousal” (+6). High arousal was described with excitement, fear, and anger, whereas low arousal was depicted as relaxation, boredom, and calmness. The FAS has been used in combination with physical activity in various studies and showed acceptable convergent validity for the English (r = 0.45 to 0.70) and the German version (r = 0.50 to 0.62) [46,48,49].

##### Positive and Negative Affect Schedule

The German version of the Positive and Negative Affect Schedule (PANAS) [52] was used in the present study [53]. It consists of twenty adjectives that describe sensations and emotions. Therefrom, two subscales can be derived: 10 adjectives inquire about positive affect (Pos. A.; e.g., alert, determined, attentive) and 10 adjectives about negative affect (NA; e.g., hostile, nervous, ashamed). The Likert rating scale consists of 5 points from “very slightly or not at all” (+1) to “extremely” (+5). Different timeframes were used for the PANAS in the past [52], but in this study, the question was related to the moment (right now). The validity in terms of content and construct of the PANAS was demonstrated in development samples; internal consistency for the current application is r = 0.86 to 0.93 [54]. The PANAS has been used frequently to assess affective responses during physical activity in adults and children [40,46].

#### 2.4.2. State Anxiety

State anxiety was evaluated by the German version of the State-Trait Anxiety Inventory (STAI) from Laux and colleagues [55]. This self-rating scale is used in the clinical field for anxiety and stress research and distinguishes between state and trait anxiety [56]. In this study, state anxiety was assessed by a scale of 10 anxiety-present and ten anxiety-absent items, which consisted of short, concise sentences, such as “I am relaxed”. On a 4-point response scale from “not at all” (+1) to “very much” (+4), patients were asked to indicate how much this statement applied to them at this moment. Convergent validity was demonstrated (r = 0.73 to 0.90), and internal reliability was about Cronbach’s α = 0.90 [55]. The STAI has often been used in conjunction with physical activity, as shown by the meta-analyses in [19,20].

#### 2.4.3. Self-Efficacy

Self-efficacy was assessed by the General Self-Efficacy Scale (SWE) from Schwarzer and Jerusalem [57]. The self-assessment procedure measures self-efficacy, which is a situation-specific construct based on Bandura’s self-efficacy concept [58]. Ten items on moment-specific optimistic self-conviction (e.g., “When resistance arises, I find ways and means to assert myself”) should be answered on a 4-point answer scale from “not true” (+1) to “true exactly” (+4). In the present study, the survey was related to perceived self-efficacy right now in the moment (pre- and postintervention). Addition of the points results in a comparable test value. Internal consistency (Cronbach’s α = 0.76 to 0.90) and criterion-related validity were demonstrated in development samples according to the authors [59]. The relationship of PA and self-efficacy was analyzed in several studies [29,60,61].

#### 2.4.4. Exercise Variables and Daily Well-Being

For comparability of the physical stress, the heart rate was recorded with the help of polar watches at the times before, during, and after the intervention. In addition, the subjective perception of physical exertion was surveyed with the rating of perceived exertion (RPE) at the same time points [62]. RPE is assessed because exercise intensity could influence affective responses [49]. The respondent had to rate perceived exertion on a scale from “no exertion” (+6) to “maximal exertion” (+20). The survey of daily well-being using the Visual Analogue Scale is used to record “bad days” in inpatient settings with particularly conspicuous answers. On a 100 millimeter line, the test persons had to mark how they assessed their daily state of well-being from “very bad” (0) to “very good” (100) [63].

### 2.5. Statistical Analyses

Data were analyzed with the statistic program, SPSS Statistics 25 (IBM, United States). All values are presented as mean values ± standard deviation; the significance level was set at *p* < 0.05. Due to the within-subject design and the normal distribution of the data, a repeated-measure analysis of variance (ANOVA) with two within-subject factors (intervention and time) could be conducted. The time factor varied between 2 and 4 time points depending on the measurement instrument. PANAS, STAI, and SWE were analyzed by a 2 (time: before, after) × 3 (experimental condition: CT, NW, SC) repeated-measure ANOVA, RPE and heart rate by a 3 (time: before, during, after) × 3 (experimental condition: CT, NW, SC) repeated-measure ANOVA, and FS and FAS by a 4 (time: before, during, after, after 180 min) × 3 (experimental condition: CT, NW, SC) repeated-measure ANOVA. One-repeated measure analysis was used to analyze potential differences of the interventions (CT, NW, SC) in daily well-being. Existing interaction effects were analyzed with a pairwise comparison using Bonferroni correction. Greenhouse–Geisser correction was applied where the assumption of sphericity was not met. Partial eta^2^ (η^2^) as effect size indicates a slight effect with 0.01, a medium effect with 0.06, and a strong effect with 0.14 [64]. 

## 3. Results

A total of 31 subjects participated in the study, whereas 21 persons completed all the three interventions and questionnaires. The high dropout rate (*n* = 10) was due to the outbreak of the coronavirus pandemic. All 10 patients had to leave earlier, and therefore, not all data of the three interventions could be collected. The average age of the final sample was 32 ± 12.2 years, and the age ranged from 18 to 57 years. A total of 13 women (61.9%) and 8 men (38.1%), who were in inpatient treatment, participated voluntarily in this study. The main diagnoses of the sample were obsessive–compulsive disorder (ICD F42; 57.1%), depression (ICD F32; 33.3%), and anxiety disorder (ICD F40, F41; 9.5%). In addition, patients named the following as further diagnoses: obsessive–compulsive disorder (ICD F42; 15%), depression (ICD F32; 30%), anxiety disorder (ICD F40, F41; 30%), eating disorder (ICD F50; 5%), post-traumatic stress disorder (ICD F43.1; 5%), and somatoform disorder (ICD F45; 5%).

### 3.1. Affective Responses

The positive affect (Pos. A.) according to the PANAS revealed significant main effects of time and intervention and a significant interaction effect (intervention × time) (Table 1). Post hoc analysis showed that all the three interventions and the two time points were significantly different. Simple contrasts revealed a significantly higher Pos. A. in CT compared with NW, *p* = 0.001, and SC, *p* < 0.001. Pos. A. changed by +0.84 points between pretest and post-test in CT, by +0.52 points in NW, and by +0.09 points in SC.

In the negative affect (NA) considering the Greenhouse–Geisser correction, the main effect of time was statistically highly significant, whereas the interventions did not differ significantly. Additionally, there was no significant interaction effect (Table 1). The NA of all the three interventions decreased significantly: in CT by −0.63 points between pre- and post-test, by −0.51 points in NW, and by −0.27 points in SC. 

For both FS and FAS, the main effects of intervention and time and the interaction effect were significant (Figure 3). Affective valence showed a significant difference in all the three interventions over time, F(3,60) = 16.504, *p* < 0.001, η^2^ = 0.452. The interventions differed significantly, F(2,40) = 5.996, *p* = 0.005, η^2^ = 0.231. Pairwise comparison showed no significant difference between CT and SC, *p* = 0.053, whereas NW and SC differed significantly, *p* = 0.009. The time effect showed a significant difference between the pretest (t_0_) to the test during intervention (t_1_), *p* = 0.001; post-test (t_2_), *p* < 0.001; and test 180 min after the intervention (t_3_), *p* = 0.017. Perceived activation showed a significant increase in all the three interventions over time, F(3,60) = 3.367, *p* = 0.024, η^2^ = 0.144. Additionally, the three interventions differed significantly, F(2,40) = 7.677, *p* = 0.002, η^2^ = 0.277. In the case of the FAS, there was a significant difference between CT and SC, *p* = 0.012, as well as the CT and NW, *p* = 0.001. Regarding the time effect, there was a significant difference between the pretest (t_0_) and test 180 min after the intervention (t_3_), *p* = 0.029, and between the test during the intervention (t_1_) and test 180 min after the intervention (t_3_), *p* = 0.036. However, the comparison of the three interventions after 180 min using single-factor analysis of variance with repeated measures was not significant, *p* = 0.195. Values of affective valence and perceived activation were collected over 4 time points and outlined using the circumplex model (see Figure 3).

### 3.2. State Anxiety

State anxiety was recorded at two points in time. Both the interventions and the timing differed significantly. The interaction effect was also significant (Table 1). The highest difference in mean value occurred in CT. There was a significant difference between NW and SC, *p* = 0.035, but no significant difference between CT and NW, *p* = 1.0, or SC, *p* = 0.081.

### 3.3. Self-Efficacy

Self-efficacy was assessed pre- and post-intervention, and significant differences between the interventions and times as well as significant interaction effects are shown (Table 1). CT differed significantly not only from SC, *p* = 0.003, but also from NW, *p* = 0.017. The largest mean difference was in CT.

### 3.4. Exercise Variables and Daily Well-Being

Physical stress was measured by the heart rate and the rating of perceived exertion at 3 time points (Table 2). In addition, the test persons were initially asked to indicate their daily well-being on a VAS. Both objectively and subjectively measured exertions showed significant time × intervention effects. CT showed the highest increase in heart rate and perceived exertion compared with NW and SC.

Daily well-being measured by the VAS scores with an average between 46.45 and 49.7 did not differ significantly between the experimental group conditions (F(2,38) = 0.182, *p* = 0.834, η^2^ = 0.009).

## 4. Discussion

### 4.1. Main Findings

This study aimed to investigate the influence of a therapeutic climbing intervention on affective responses, anxiety, and self-efficacy in psychosomatic patients in comparison with Nordic walking intervention and a sedentary control condition. Significant time by intervention interaction effects showed that CT evoked higher affective responses in positive affect and self-efficacy compared with both control interventions. Therefore, the hypothesis that therapeutic climbing has a greater impact on affective responses, state anxiety, and perceived self-efficacy than an endurance sport and a sedentary control condition can only be partly accepted. For negative affect and state anxiety, there is a temporal effect for all the three interventions, but the CT does not differ significantly from the other two conditions. In addition, there is no significant difference in affective responses between the three conditions 180 min after the intervention.

### 4.2. Affective Responses

Changes in affective responses were seen between the interventions over time. When looking at positive affect, it could be seen that it increased over time in the CT compared with the SC and differed significantly from NW, which also increased over time. There were no group condition differences in negative affect between all the three interventions, but there was a main effect for time, with negative affect decreasing over time for both CT and NW. An increase in affective valence can lead to better therapy adherence due to greater motivation [46,65,66]. The changes in affect can also be explained by the various experiential qualities of climbing. Kleinstäuber and colleagues suggested that CT may contribute to emotion regulation, which could explain the reduction in negative affect [40]. NW as an endurance sport also showed this emotion regulation effect in the present study and in previous research [45]. However, the change is not as pronounced as in CT. The authors explained this with the high demand on concentration and coordination during climbing [40]. They assumed that the immediate experience of different feelings on the wall and a quick goal achievement, for example, climbing through a route, can be associated with short-term emotion regulation effects [40]. In contrast, the range of emotions that can be experienced and the feeling of goal achievement are less pronounced with NW, which is why the influence on emotions could also be lower. However, other aspects of physical activity, which are present in CT and NW, such as physical activation and working up a sweat, could influence positive and negative affect [67].

In the present study, affective responses differed significantly between the interventions over time. The circumplex model, consisting of the two scales FS and FAS, can be used to interpret affective responses and activation [68]. The optimal affective state during exercise is the upper-right quadrant, characterized by high activation and high affective valence, which is related to power and energy (see Figure 3). In the present study, affective valence and perceived activation were in the quadrant for high activation pleasure and energy for the CT intervention but not for the control interventions. Williams and colleagues concluded from their study that a positive change in affective valence during exercise leads to higher future exercise [66]. An increase of about 30 min of moderately intense physical activity at 6 and 12 months after the intervention was associated with an increase in one unit on the FS [66]. In both exercise interventions (CT and NW) in the present study, an increase of about one unit on the FS was reported between before and during the intervention. This can also be seen as an indicator for increased adherence to therapy [65,66]. In addition, various experiential qualities of CT could contribute to increasing affective responses: social interaction (joy, communication, group membership), mindfulness (being in the moment, no thoughts, no ruminations, and no compulsions), experiencing success and learning coping strategies, experiencing joy and flow, and increasing self-esteem and self-efficacy [33,34,35,40,41,69]. How and to what extent these aspects influence the affective valence have not been studied in detail yet. The increase in activation could be explained by the highly challenging nature of climbing and higher anxiety or nervousness in CT [46]. Dealing with the potential of falling could lead to long-term changes in self-efficacy and self-esteem [33,34,40]. Although affective responses differed between the interventions immediately after the intervention, this difference between CT and NW no longer existed after 180 min. It seems that CT had no greater long-term effect on affective responses than the NW. 

Looking at the rating of perceived exertion (RPE), a relationship between physical strain and influence on changes in positive and negative affect can be suspected. RPE in NW was lower than in CT during the intervention, and the changes in positive and negative affect were smaller in NW compared with CT. In children and adolescents, there was no significant difference in positive and negative affects between climbing, swimming (as an endurance sport), and occupational therapy [46]. For all the three interventions, the authors demonstrated an increase in positive affect and a reduction in negative affect. In comparison with the results of Kleinstäuber and colleagues and those of the present study, it could be assumed that CT has a stronger effect on affective responses in adults than in children and adolescents [40].

### 4.3. State Anxiety

State anxiety showed a significant interaction effect between intervention and time. However, there was no significant difference between CT and NW or SC—only NW and SC differed significantly. Although there was no interaction effect of CT with the other interventions, state anxiety also decreased significantly over time with CT. Therefore, it could be assumed that physical activity per se can be anxiety reducing. This is in line with results by Gallotta and colleagues, which compared supervised rock-climbing training and fitness training in adults over 3 months [70]. After 3 months, anxiety reduced significantly in both groups. The authors suggested that the decrease in state anxiety could be explained by the distraction hypothesis [71]. Other researchers also compared an aerobic exercise with a climbing intervention and found anxiety-reducing effects in both interventions [34]. A large number of studies also led to this conclusion regarding CT: climbing reduces anxiety and can thus contribute to emotional stability and has a positive effect on the treatment of depression [72]. Regarding the question of whether this anxiety-reducing effect relates predominantly to climbing or is due to physical activity in general, the results of the present study indicated that physical activity itself leads to a reduction in anxiety. However, the results of the present study could not confirm the statement that climbing has a greater anxiety-reducing effect than a fitness program, because both exercise interventions had similar effects over time [34]. The reduction of anxiety especially in CT can be accounted for by the presence of anxiety in climbing. The emotion of fear is omnipresent in climbing, and the combination of confronting it and working out solution strategies regarding how to deal with it has a positive effect on coping with anxiety [33,35,46]. Another reason for the successful anxiety reduction in CT could be the ‘leaning towards the exposure procedure’ [73]. Getting confronted with physiological sensations of exercise (sweat, rapid heartbeat), which are associated with anxiety and panic, could increase the tolerance for such symptoms and facilitate the handling of them [73]. The mastering of the situation could be associated with self-efficacy and internal locus of control [35]. The reduction of fear can probably be attributed not only to the confrontation with it but also to the strengthening of self-esteem and other safety-giving aspects of climbing, such as active and passive coping strategies [33,34,35]. By confronting anxiety symptoms during physical activity, patients with mental health disorders might learn to deal with anxiety naturally and can cognitively reclassify and process the new experiences. In combination with other aspects (social, self-effective), this could improve the psychological symptomatology [73,74].

### 4.4. Self-Efficacy

Self-efficacy showed a significant time by intervention effect with increased moment-specific self-efficacy after CT compared with both NW and SC—therefore, CT may increase participants’ beliefs in their capabilities to perform target-oriented behaviors in general. Various studies have already evaluated self-efficacy in climbing and bouldering and have also postulated improvements in perceived self-efficacy over time [33,34,35,38,39,40,41]. In addition to this acute effect of the present study, two researchers from the same group evaluated that self-efficacy increased more through a CT intervention compared with a general sports program [34,38]. Increased self-efficacy through therapeutic climbing might be explained by the following climbing-specific components that are not present in NW or in an aerobic fitness program [34]: Mastering a climbing route could result in a sense of achievement and pride in having accomplished something on its own [33,41]. Learning coping strategies, which lead directly to success on the wall, and repeating them leads to an increase in self-esteem and support self-efficacy [33,39]. Additionally, dealing with various emotions (positive and negative) during climbing and the relief of being able to deal with them can lead to pride and an increase in self-efficacy [40]. It could be assumed that also social and symbolic components of CT (social relationships between the members, communication, being part of a group, and goal achievement by mastering a climbing route) improve self-efficacy [33,40]. According to social cognitive theory by Bandura, greater self-efficacy leads to more confident handling of threatening or stressful situations [73]. Self-efficacy as a situation-specific form of self-confidence correlates naturally with health behavior and influences psychological well-being [23,75]. Increasing self-efficacy through CT and NW enables people with mental health disorders to cope better and should be explored even more [33,38,76].

### 4.5. Strengths and Limitations

The present study is, to the best of the authors’ knowledge, the first to investigate, as a randomized clinical trial with a within-subjects design, the effects of rope climbing therapy compared with two control interventions (exercise and sedentary control condition) in an inpatient setting. Not only patients with depression but also anxiety and obsessive–compulsive disorder patients were included in the study. Data collection in the clinical inpatient setting is rare, but CT is becoming more common in psychiatric treatment [76]. The present results contribute to a greater understanding of the effects of exercise and specifically climbing therapy in the clinical treatment context. Comparison with an SC intervention instead of a resting intervention establishes the relation to everyday (clinical) life. Daily well-being did not differ between the experimental conditions. Thus, it can be assumed that daily well-being did not influence any intervention outcomes. 

The main limitation of the pilot study can be considered the relatively small sample size, which further did not allow us to analyze differences between gender and disease pattern. Additionally, gender ratio and the ratio of disease patterns were not balanced. Another limitation is data collection with the same experiment manager: the bias due to different experimenters is reduced, but it cannot be excluded that the subject’s affection for this person led to biased data. Another bias could be the environmental setting: CT was conducted indoors, whereas NW took place outdoors, and for SC, the subjects could choose whether they stay inside or go out. Frühauf and colleagues [14] suggested that an outdoor setting could influence affective responses in people with mental health disorders. Group cohesion is also a limitation factor because climbing took place in a steady group, whereas the NW group differed in group members. In addition, the NW group consisted of almost twice as many participants as in the CT. Further data collection during intervention can be mentioned critically because it was not possible to answer the questionnaires accurately at half of the time, and the participants answered the questions while they had a short relaxation time (they stood or sat while answering). Another weakness of the study is the different intensity and load in the exercise conditions. As this is a pilot study to evaluate typically practiced forms of exercise in comparison with climbing therapy, this weakness was accepted in order to not create too many difficulties in everyday clinical practice.

## 5. Conclusions

Exercise therapy led to positive changes in affective responses, self-efficacy, and anxiety with therapeutic climbing showing additional benefits related to the improvement of self-efficacy and affective responses compared with Nordic walking and a sedentary control condition in a psychiatric inpatient setting. The high effect sizes found in the present study for CT should be replicated in further clinical trial settings. Especially, comparisons with exercise interventions of similar intensity should be undertaken.

## Figures and Tables

**Figure 1 ijerph-19-06767-f001:**
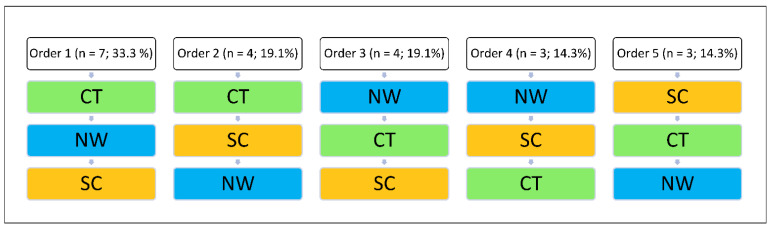
Five different sequences for the three interventions climbing therapy (CT), Nordic walking, (NW) and sedentary control condition (SC).

**Figure 2 ijerph-19-06767-f002:**
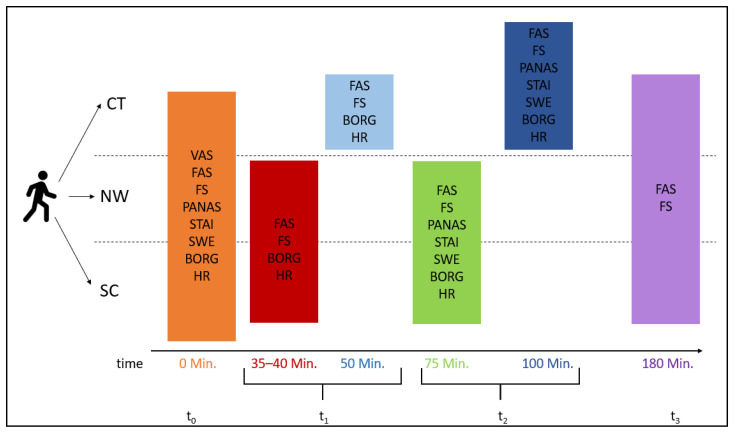
Overview of tests with chronological order before (t_0_), during (t_1_), and after (t_2_) and 180 min after the intervention (t_3_), subdivided for therapeutic climbing (CT), Nordic walking (NW), and sedentary control condition (SC). Measurement instruments: Visual Analogue Scale (VAS), Felt Arousal Scale (FAS), Feeling Scale (FS), Positive and Negative Affect Schedule (PANAS), State-Trait Anxiety Inventory (STAI), General Self-Efficacy Scale (SWE), Borg Scale (BORG), and heart rate (HR).

**Figure 3 ijerph-19-06767-f003:**
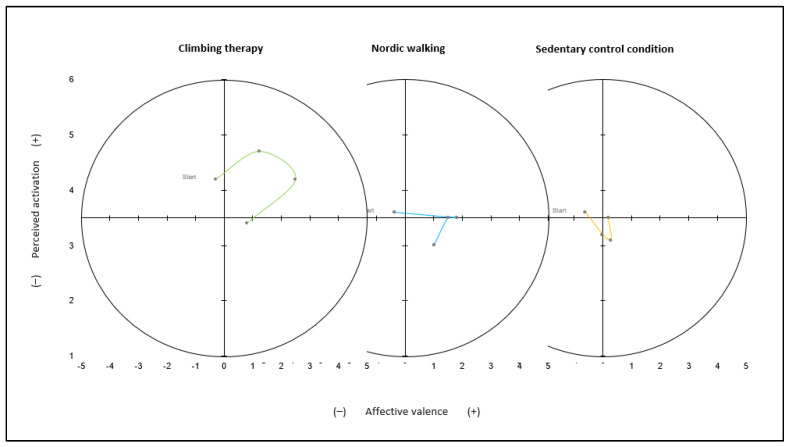
Mean affective valence and perceived activation over time for all the three interventions and 4 points in time.

**Table 1 ijerph-19-06767-t001:** Results of positive and negative affect, state anxiety and self-efficacy for the three interventions, before-after-comparison.

Variable	Intervention	Pre		Post		Time			Intervention			Time*Intervention		
		M	(SD)	M	(SD)	F (1,20)	*p*	η^2^	F (2,40)	*p*	η^2^	F (2,40)	*p*	η^2^
Positive Affect	CT	2.79	(0.7)	3.63	(0.9)	30.014	<0.001 ***	0.60	26.180	<0.001 ***	0.567	8.957	0.001 **	0.309
	NW	2.46	(0.6)	2.98	(0.7)									
	SC	2.17	(0.6)	2.26	(0.8)									
Negative Affect	CT	2.24	(0.7)	1.61	(0.7)	51.512	<0.001 ***	0.72	2.072	0.154	0.094	2.626	0.089	0.116
	NW	2.09	(1.0)	1.58	(0.8)									
	SC	2.2	(1.0)	1.93	(1.0)									
State anxiety	CT	53.90	(10.9)	43.95	(11.0)	27.128	<0.001 ***	0.576	5.020	0.011 *	0.201	6.603	0.003 **	0.248
	NW	53.29	(13.4)	44.86	(10.6)									
	SC	54.38	(12.8)	52.33	(13.0)									
Self-efficacy	CT	22.29	(5.9)	26.76	(6.1)	27.200	<0.001 ***	0.576	8.496	0.001 **	0.298	6.046	0.005 **	0.232
	NW	22.14	(7.13)	23.71	(6.6)									
	SC	21.67	(6.3)	22.67	(7.1)									

Note. CT = climbing therapy, NW = Nordic walking, SC = sedentary control condition; * *p* < 0.05, ** *p* < 0.01, *** *p* < 0.001.

**Table 2 ijerph-19-06767-t002:** Heart rate and RPE results of the three interventions at 3 measurement time points.

Variable	Intervention	t_0_		t_1_		t_2_		Time			Intervention			Time*Intervention		
		M	(SD)	M	(SD)	M	(SD)	F (2,40)	*p*	η^2^	F (2,40)	*p*	η^2^	F (4,80)	*p*	η^2^
Heart rate (bpm)	CT	90.5	(14.8)	105.5	(22.1)	102.7	(18.0)	7.837	0.002 **	0.359	20.004	<0.001 ***	0.588	2.970	0.027 *	0.175
	NW	85.7	(15.8)	93.0	(15.6)	85.9	(10.9)									
	SC	76.3	(7.7)	75.0	(11.1)	78.9	(11.0)									
RPE	CT	5.95	(3.3)	9.76	(2.7)	9.48	(2.7)	11.550	<0.001 ***	0.366	22.404	<0.001 ***	0.528	12.041	<0.001 ***	0.376
	NW	6.0	(3.6)	7.57	(1.9)	7.38	(2.1)									
	SC	5.1	(3.6)	4.48	(3.1)	4.57	(3.2)									

Note. CT = climbing therapy, NW = Nordic walking, SC = sedentary control condition; * *p* < 0.05, ** *p* < 0.01, *** *p* < 0.001.

## Data Availability

The data are not publicly available due to ethical considerations on preserving the anonymity of study participants.

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
