# Peer review of "A Comparison of Acute Effects of Climbing Therapy with Nordic Walking for Inpatient Adults with Mental Health Disorder: A Clinical Pilot Trial"

_ijerph, 2022, doi:10.3390/ijerph19116767_

Round 1
Reviewer 1 Report
The study aimed to compare the acute effects of a therapeutic climbing intervention on affective responses, anxiety, and self-efficacy with Nordic Walking and a sedentary control condition in people with a variety of mental disorders.
General comments:
The manuscript is clear, well written, and shows a well-structured manner. The topic is original and relevant to the field of public health and exercise. The design is appropriate and therefore the results and conclusions are consistent. My Congratulations on one of the best graphic designs I have ever seen in a scientific paper.
- Specific comments
Several limitations of the study as the small size of the sample and the short term of the interventions must be taken into account. Following that, probably the title of the study must include any terms such as "pilot study"; "preliminary study" or similar.
Author Response
First of all, we would like to thank the editor and reviewers for their constructive feedback and valuable comments to the manuscript and the opportunity to revise our paper. We believe that the comments helped improving the manuscript. In the notes below, we will address the comments of the reviewers. Please find the comments of the reviewers in "blue" and the author´s responses in “black”.
General comments:
The manuscript is clear, well written, and shows a well-structured manner. The topic is original and relevant to the field of public health and exercise. The design is appropriate and therefore the results and conclusions are consistent. My Congratulations on one of the best graphic designs I have ever seen in a scientific paper.
Thank you for this positive feedback.
Specific comments
Several limitations of the study as the small size of the sample and the short term of the interventions must be taken into account. Following that, probably the title of the study must include any terms such as "pilot study"; "preliminary study" or similar.
Thank you for your comment. We are aware of those limitations. We have addressed the small sample size in the limitations. However, measuring the short-term effects of the activity was the aim of the study, but shortcomings of measuring acute effects are discussed in the paper. According to your suggestion, we have changed the title and included pilot trial which now reads as following: ‘A comparison on acute effects of climbing therapy to Nordic Walking for inpatient adults with mental health disorder: A clinical pilot trial’.
Reviewer 2 Report
The text shows a study on the psychological effects of activation through physical exercise with two conditions (climbing, walking) versus sedentary control. The effects are measured immediately, and participants are counterbalanced by the three conditions. The study is well developed and justified, the methodology is appropriate, as well as the data analysis performed. However, it presents some problems that could improve the final article.
- The title needs to be smaller and clearer, something like: “A comparison of climbing therapy to Nordic walking for impatient adults with mental health disorders”
- It should be specified in the abstract that the very immediate psychological effects of physical exercise are measured.
- The review only includes studies on physical exercise, and does not include psychological studies showing Behavioural Activation as an effective and efficient therapy for depressive and emotional problems. Such therapies have long demonstrated the effectiveness of activation/exercise/social contact for mental health. It is not only physical exercise, but any activity that reinforce the individual in his or her natural environment. In the discussion, the authors talk about these effects (429-432), but they have not taken them into account in the review.
- Regarding the design, although it is the usual nomenclature of within-subjects, "within-groups with repeated measures and counterbalancing of participants" should be used. It is not comparing participants one by one, but as a whole group. It is not an N=1 design.
- It should be better clarified the procedure, how many days of exercise, walking and sedentary lifestyle?. The sequence seems too rapid, there are sure interactions or cumulative effects on the results. This should be commented on in the discussion.
- There is little time between tests, there is a memory and/or repetition effect. Questionnaires cannot be administered repeatedly on the same day, this invalidates their reliability. This should be commented on in the discussion.
- In the description of the instruments, reliability data and whether there are cut-off points or clinical criteria in them should be given.
- The conclusions in "main findings" (320-327) do not correspond exactly to the analysis of the results (in negative affectivity, anxiety, and comparisons at 180 minutes there are no significant effects).
- Repeated measurement will always result in an observed effect, it is an artefact of internal validity. It should fit the description to the data, and be discussed afterwards.
- The references should be revised in detail, there are errors in capital letters in the names of journals and authors. The DOI of the articles should be included.
Reviewer 3 Report
I enjoyed this study about the acute effects of physical activities on mood and anxiety. I agree with the limitations of the study as you mentioned. To draw the firm conclusions from your hypothesis, I would like to see the study with bigger sample.
Author Response
First of all, we would like to thank the editor and reviewers for their constructive feedback and valuable comments to the manuscript and the opportunity to revise our paper. We believe that the comments helped improving the manuscript. In the notes below, we will address the comments of the reviewers. Please find the comments of the reviewers in “blue” and the author´s responses in “black”.
I enjoyed this study about the acute effects of physical activities on mood and anxiety. I agree with the limitations of the study as you mentioned. To draw the firm conclusions from your hypothesis, I would like to see the study with bigger sample.
Thank you for this positive feedback. A follow-up study with a larger sample is planned for the future.
Reviewer 4 Report
This is an interesting study examining an important clinical issue. I have some suggestions for the authors to improve their manuscript.
- Error: Line 44: [23]:
- Error: Line 48: self-esteem [25, 26].
- Line 64: “Researchers found that bouldering psychotherapy is as effective as cognitive behavioural therapy in terms of both short-term and long-term effects [35].” Additional explanation would be helpful for the readers to understand its meaning.
- Line 74: “So far, only the long-term and not the short term effects immediately after the intervention have been recorded, which is why the present study aims to measure the acute changes in CT in comparison to an endurance sport and a control group.” Why the acute effect should be examined should be explained.
- Line 76: “Given the higher economic effort of CT compared to other group therapies (small groups, equipment, safety)” I do not understand why CT has the advantages on small groups, equipment, safety compared with other PA such as aerobic exercise?
- Line 124-130: the letters was smaller than other parts.
- Line 159: “Nordic Walking is a recommended aerobic exercise for patients with mental disorders [43]” and Line 160 “An improvement of the current mental state and mood-lifting effects have been observed frequently in depressed persons [17, 44, 45]” should be mover to Introduction section.
Author Response
First of all, we would like to thank the editor and reviewers for their constructive feedback and valuable comments to the manuscript and the opportunity to revise our paper. We believe that the comments helped improving the manuscript. In the notes below, we will address the comments of the reviewers. Please find the comments of the reviewers in “blue” and the author´s responses in “black”.
This is an interesting study examining an important clinical issue. I have some suggestions for the authors to improve their manuscript.
Error: Line 44: [23]:
Error: Line 48: self-esteem [25, 26].
Thank you for noticing, we’ve corrected both mistakes.
Line 64: “Researchers found that bouldering psychotherapy is as effective as cognitive behavioural therapy in terms of both short-term and long-term effects [35].” Additional explanation would be helpful for the readers to understand its meaning.
Thank you for this comment. To make the sentence more understandable, we revised the sentence which no reads as following: ‘In a recent study, Luttenberger et al. [35] found bouldering psychotherapy to be as effective as cognitive behavioral therapy. Those results were seen after 10 weeks of intervention and maintained up to one year after intervention [35].’ (Lines 69-72).
Line 74: “So far, only the long-term and not the short-term effects immediately after the intervention have been recorded, which is why the present study aims to measure the acute changes in CT in comparison to an endurance sport and a control group.” Why the acute effect should be examined should be explained.
Thank you for this comment and an important point. To clarify the argument, we added a short explanation which reads as the following: ‘So far, only the long-term and not the acute effects immediately after the intervention have been evaluated, which is why the present study aims to assess acute changes of psychological variables through CT in comparison to an aerobic PA and a control group. Acute effects are of particular interest in the inpatient setting, as the change can be predominantly attributed to the intervention in question.’ For better understanding, we changed the order of the argumentation and thereby added the sentence to the thought process: ‘The importance of affective responses to treatment participation and PA behavior also reinforces the need to examine short-term effects [39, 48].’ (Lines 101-102).
Line 76: “Given the higher economic effort of CT compared to other group therapies (small groups, equipment, safety)” I do not understand why CT has the advantages on small groups, equipment, safety compared with other PA such as aerobic exercise?
Thank you for asking. This is a misunderstanding. CT means more effort compared to other sports interventions such as Nordic walking, because the groups have to be smaller, the staff has to be trained and the material (climbing wall, belay devices, harnesses) is more expensive. Therefore, this is seen as a disadvantage rather than an advantage compared to the other interventions. We changed the sentence and hope that it is now more understandable: ‘Since CT is characterized by a higher economic cost compared to most PA group therapies (e.g., infrastructure and equipment, high staff to patient ratio) Frühauf and colleagues call for a randomized-controlled trial with the three interventions climbing, aerobic exercise, and social contact group in a homogeneous group of patients [34].’ (Lines 89-93).
Line 124-130: the letters was smaller than other parts.
Thank you for noticing. We changed it accordingly to the correct size.
Line 159: “Nordic Walking is a recommended aerobic exercise for patients with mental disorders [43]” and Line 160 “An improvement of the current mental state and mood-lifting effects have been observed frequently in depressed persons [17, 44, 45]” should be mover to Introduction section.
Thank you for this comment. We moved the two sentences to the introduction section and embed them in the context.
Reviewer 5 Report
The manuscript entitled "Acute Effects of Climbing Therapy in Comparison to Nordic Walking on Affective Responses, State Anxiety, and Self-Efficacy in Adults With Mental Health Disorders During Inpatient Treatment: a Pragmatic Practice-Based Clinical Trial" is an interesting and important work for behavioral medicine and health psychology areas of science. In particular, climbing therapy has a beneficial effect on the mental health of patients with depression, OCD, and anxiety disorders. Although the manuscript is clear and well-written, some issues need to be resolved before reconsidering this paper to publication.
- The main concern is a small sample size, which is a big problem for self-report questionnaires and statistical tests (a minimal group considering for comparison with another sample should count a minimum 30 people). Although the paper is a preliminary study, the small sample size should be discussed as a main weakness of this study, which decrease confidence in the results.
- The second problem is not balanced gender samples, and lack of gender comparisons in statistical analyses. It is well-known that women and men differ in all emotion-related variables, as well as in depression or anxiety scores, when self-reported measures are performed. This fact was not considered in the study, and also not discussed in the limitation section of the manuscript.
- The third weakness of this study is that the sample is not homogeneous in terms of mental disorders. It is possible that three treatment conditions (ST, NW, and SCC) have various impacts on the mental state of patients depending on their main diagnosis and comorbidities. This was not controlled or discussed as a limitation.
- Also, repeated self-report measures are a source of concern. When people answer the same questions many times with a short duration, they can automatically select the responses. It can be another source of bias and should be discussed and included in a limitation section.
- The reliability of each scale used in the study should be reported for previous German validation studies, as well as for the present sample in the current study, in each point of time (t0, t1, t2, t3).
- What type of post-hoc test was performed to find significant differences between groups and conditions in ANOVA?
- Was used any p-value correction to avoid bias related to multiple comparisons (e.g., Bonferroni correction)?
- There is inconsistency in reporting the results in text. Sometimes the authors reported F, df and p for post-hoc comparisons, while sometimes not (e.g., see sections 3.2., 3.3., 3.4.). It should be improved.
Author Response
First of all, we would like to thank the editor and reviewers for their constructive feedback and valuable comments to the manuscript and the opportunity to revise our paper. We believe that the comments helped improving the manuscript. In the notes below, we will address the comments of the reviewers. Please find the comments of the reviewers in “blue” and the author´s responses in “black”.
The manuscript entitled "Acute Effects of Climbing Therapy in Comparison to Nordic Walking on Affective Responses, State Anxiety, and Self-Efficacy in Adults With Mental Health Disorders During Inpatient Treatment: a Pragmatic Practice-Based Clinical Trial" is an interesting and important work for behavioral medicine and health psychology areas of science. In particular, climbing therapy has a beneficial effect on the mental health of patients with depression, OCD, and anxiety disorders. Although the manuscript is clear and well-written, some issues need to be resolved before reconsidering this paper to publication.
- The main concern is a small sample size, which is a big problem for self-report questionnaires and statistical tests (a minimal group considering for comparison with another sample should count a minimum 30 people). Although the paper is a preliminary study, the small sample size should be discussed as a main weakness of this study, which decrease confidence in the results.
Thank you for your comment. We are aware of the small sample size and agree with the reviewer. However, since we did not conduct a single group repeated measure design but used a within-subject design, where every patient took part in every condition, the sample size is not as small as it seems. Comparable sample sizes have been reported previously (Kleinstäuber et al., 2017, DOI: 10.2147/PRBM.S143830; Mazzoni et al., 2009, DOI: 10.1123/apaq.26.3.259).
However, we addressed this advice in the manuscript and it now reads as following as the first sentence of limitations: ‘The main limitation of the pilot study can be considered the relatively small sample size which further did not allow to analyze differences between gender and disease pattern. Also gender ratio and the ratio of disease patterns were not balanced.’ (Lines 494-496).
2. The second problem is not balanced gender samples, and lack of gender comparisons in statistical analyses. It is well-known that women and men differ in all emotion-related variables, as well as in depression or anxiety scores, when self-reported measures are performed. This fact was not considered in the study, and also not discussed in the limitation section of the manuscript.
Thank you for this feedback. As a pilot study, it was not possible for us to ensure a balanced sample in regard to gender and disease pattern. However, due to the within-subject design, changes are calculated individually on the person and thus, sociodemographic differences within the sample weigh less in the analyses. We could have integrated gender or disease pattern as a covariate in the ANOVA, however, decided against it due to the small sample size. We’ve integrated this aspect in the limitations, and it now reads as following: ‘The main limitation of the pilot study can be considered the relatively small sample size which further did not allow to analyze differences between gender and disease pattern. Also gender ratio and the ratio of disease patterns were not balanced.’ (Lines 494-496).
- The third weakness of this study is that the sample is not homogeneous in terms of mental disorders. It is possible that three treatment conditions (ST, NW, and SCC) have various impacts on the mental state of patients depending on their main diagnosis and comorbidities. This was not controlled or discussed as a limitation.
Thank you for this comment. As explained in our previous comment, we could not consider the even ratio of the disease patterns. However, we do not see this as a limitation mainly due to the fact that the intervention is a pragmatic clinical trial and should be feasible for a wide variety of psychiatric disorders. As mentioned in the previous comment, we have integrated this aspect in the limitation section.
In a follow-up study with a larger sample, the different disease patterns will be taken into account in the analysis of the results.
4. Also, repeated self-report measures are a source of concern. When people answer the same questions many times with a short duration, they can automatically select the responses. It can be another source of bias and should be discussed and included in a limitation section.
Thank you for pointing out this aspect. We agree with the reviewer that memory and repetition effects could occur, however, we believe that this has not affected our results due to the following reasons:
a) Memory effects are more common in questionnaires assessing cognitive changes but are less mentioned in research about acute changes in mood. The questionnaires we used were designed for detecting acute changes and are used and recommended by various researchers for repeated measurements over short time intervals (Ekkekakis & Petruzzello, 2002, doi: 10.1016/S1469-0292(01)00028-0; Ekkekakis et al., 2008, doi: 10.1016/j.psychsport.2007.04.004).
b) In order to relate the results to the intervention, a control condition was integrated which should neglect memory and repetition effects through the questionnaires.
5. The reliability of each scale used in the study should be reported for previous German validation studies, as well as for the present sample in the current study, in each point of time (t0, t1, t2, t3).
Thank you for this comment. We have revised the description of the measurement instruments and inserted the quality criteria.
6. What type of post-hoc test was performed to find significant differences between groups and conditions in ANOVA?
Thank you for noticing. Post-hoc analysis was performed as pairwise comparisons with Bonferroni correction using SPSS. We added this information in the manuscript, and it now reads as following: ‘Existing interaction effects were analyzed with a pairwise comparison, using Bonferroni correction.’ (Lines 280-281).
7. Was used any p-value correction to avoid bias related to multiple comparisons (e.g., Bonferroni correction)?
Thank you for asking. Yes, Bonferroni correction was used. We added this detail in the manuscript in the statistical analyses section and it now reads as following: ‘Existing interaction effects were analyzed with a pairwise comparison, using Bonferroni correction.’ (Lines 280-281).
8. There is inconsistency in reporting the results in text. Sometimes the authors reported F, df and p for post-hoc comparisons, while sometimes not (e.g., see sections 3.2., 3.3., 3.4.). It should be improved.
Thank you for this comment. For better clarity, no values from the tables have been mentioned again in the text. The p-values and the partial Eta squared can be found in the tables. Since the results of the Feeling Scale and the Felt Arousal Scale are only presented graphically, the statistical values have been fully integrated into the text. This procedure for better readability was also carried out in other studies (e.g., Frühauf et al., 2020, DOI: 10.1016/j.psychres.2020.113245).
Round 2
Reviewer 5 Report
Thank you for the exhaustive answers to my previous comment. Unfortunately, significant changes were not included in the current version of the manuscript. A way of reporting statistics is incoherent and chaotic now. The most important is to report all statistics, such as M, sd, F, df, p and η2 for each measurement. Unfortunately, no more statistics were included in the text or tables from the previous version of the manuscript. Please see the APA standards guideline to see how to arrange a table for the repeated-measures ANOVA, to be transparent and to include all the required statistics. Please add to Table 1 and Table 2 missing statistics, such as F and df for effect of time, and intervention, as well as for interaction effect. Furthermore, there are some symbols ("a" and "*"), which are not explained in the notes below table 1 and table 2. Also, three decimals and zero before point should be shown for each number in table 1 and table 2.
Author Response
Thank you for your feedback. We have followed up on your constructive criticism and added the missing values to the tables now. We have adopted the specification of the F-statistic with three decimal numbers, but for reasons of clarity and in line with the latest APA guidelines we have decided against the zero before point at the P-value. We hope that all information is now included in the tables as well as in the text.